# Does Locally Advanced Thyroid Cancer Have Different Features? Results from a Single Academic Center

**DOI:** 10.3390/jpm12020221

**Published:** 2022-02-05

**Authors:** Marco Dell’Aquila, Pietro Tralongo, Giuseppe De Ruggieri, Mariangela Curatolo, Luca Revelli, Celestino Pio Lombardi, Alfredo Pontecorvi, Guido Fadda, Luigi Maria Larocca, Marco Raffaelli, Liron Pantanowitz, Esther Diana Rossi

**Affiliations:** 1Division of Anatomic Pathology and Histology, Fondazione Policlinico Universitario “Agostino Gemelli”-IRCCS, 00168 Rome, Italy; mzrk07@gmail.com (M.D.); pietrotralongo@gmail.com (P.T.); mariangela.curatolo@guest.policlinicogemelli.it (M.C.); guido.fadda@policlinicogemelli.it (G.F.); luigimaria.larocca@policlinicogemelli.it (L.M.L.); 2Division of Endocrine Surgery, Fondazione Policlinico Universitario “Agostino Gemelli”-IRCCS, 00168 Rome, Italy; giuseppe.deruggieri@libero.it (G.D.R.); luca.revelli@policlinicogemelli.it (L.R.); celestinopio.lombardi@policlinicogemelli.it (C.P.L.); marco.raffaelli@policlinicogemelli.it (M.R.); 3Division of Endocrinology, Fondazione Policlinico Universitario “Agostino Gemelli”-IRCCS, 00168 Rome, Italy; alfredo.pontecorvi@policlinicogemelli.it; 4Department of Pathology& Clinical Labs, University of Michigan, Ann Arbor, MI 48103, USA; lironp@med.umich.edu

**Keywords:** aggressive, locally advanced, thyroid, cancer, carcinoma, recurrence, genetic alterations

## Abstract

Background: Despite the fact that the majority of thyroid cancers are indolent, 15% of patients with well-differentiated carcinoma including papillary thyroid carcinoma (PTC) present with locally advanced thyroid cancer (LATC) at diagnosis. The current study analyzes a cohort of patients with LATC focusing on their risk for local recurrence, distant metastases, and overall survival. Materials and methods: From January 2010 to December 2020, 65 patients with LATC were retrieved, including 42 cases with preoperative cytological samples. *BRAF^V600E^* and *TERT* mutations were performed on both cytology and histopathology specimens in this cohort. Results: Among the 65 cases, 42 (65%) were women. The median age was 60.1 years. Histological diagnoses included 25 (38.4%) with classic PTC and 30 (46.1%) aggressive variants of PTC, mostly tall cell variant (17 cases, 26.1%). Multifocality was seen in 33 cases (50.8%). All patients had nodal metastases. The most common site of extrathyroidal extension was the recurrent laryngeal nerve (69.2%). Staging revealed 21 cases were stage I, none were stage II, 33 were stage III, and 7 were stage IVa and 4 stage IVb. No differences were found between well and poorly/undifferentiated thyroid cancers. Conclusion: These data suggest that locally advanced thyroid cancers, including variants of PTC, exhibit a more aggressive biological course and should accordingly be more assertively managed.

## 1. Introduction

It is well-known that the incidence of thyroid cancer in developed countries is significantly increased [1,2,3,4,5,6,7,8]. The reason for this rise is ascribed largely to the detection of small thyroid cancers, mostly due to ultrasound pre-surgical evaluation. In most cases, thyroid cancers comprise well-differentiated carcinomas with an excellent prognosis. Around 13–15% of afflicted patients have locally aggressive and advanced cancers with a poor prognosis [9,10,11,12,13,14,15,16,17]. In this regard, patients presenting with locally advanced thyroid cancer (LATC) usually have an adverse prognosis and are defined by the presence of an extra thyroid extension, e.g., when the surrounding structures such as the trachea, larynx, esophagus, and main blood vessels are invaded by cancer, and in presence of ETE and ENE, regardless of histotypes [12,13,14,15,16,17,18,19,20,21]. Patients with advanced thyroid cancer often manifest with hoarseness, dysphagia, and hemoptysis. However, a small proportion (12%) of patients presenting with invasive thyroid cancer may be asymptomatic, despite having histological evidence of extrathyroidal extension (ETE).

LATC is not only associated with ETE and extranodal extension (ENE), but also certain thyroid cancer histotypes. Thyroid carcinomas that commonly behave as locally aggressive tumors include aggressive variants of PTC such as tall cell variant (TCV), columnar cell variant (CC.PTC), solid and hobnail variant, as well as poorly differentiated thyroid carcinoma (PDTC), anaplastic thyroid carcinoma (ATC), and medullary thyroid carcinoma (MTC) [12,13,14,15,16,17,18,19,20]. According to the American Thyroid Association (ATA) guidelines, aggressive variants of PTC are conferred intermediate risk, mostly due to their increased frequency of metastases, lack of avidity to radioiodine (RAI) therapy, and thus lower survival rate [21].

The current treatment modalities include appropriate surgery, radioactive iodine treatment, and external beam radiation therapy. The initial therapeutic approach to DTC consists of total thyroidectomy, which might be followed by radioactive iodine (RAI) administration, and TSH suppression in selected cases. Most authors, especially those whose therapeutic concepts include shaving at the laryngo-tracheal area or incomplete resections, advocate postoperative radioiodine therapy for differentiated tumors. [12,13,14,15,16,17,18,19,20,21] and external high-voltage therapy as supplementary ablation for aggressive forms. Furthermore, multikinase inhibitors (MKI) are drugs indicated for bulky or rapidly progressing iodine-refractory metastatic DTC that result in symptomatic or threatening uncontrolled disease not amenable to other therapies [20,21].

Some authors investigated the molecular mechanisms involved in the more aggressive behavior of a minority of thyroid cancers. The most commonly involved molecular pathway seems to be linked to the dysregulation of the mitogen-activated protein kinase (MAPK) and phosphatidylinositol-3 kinase (PI3K)/AKT signaling pathways. MAPK activation is considered to be crucial for PTC initiation, through point mutations of the *BRAF* and *RAS* genes or gene fusions of *RET/PTC* and *TRK*. On the other hand, PI3K/AKT activation is thought to be critical in FTC initiation and can be triggered by activating mutations in *RAS, PIK3CA*, and *AKT1*, as well as by inactivation of *PTEN*, which negatively regulates this pathway. TC progression and dedifferentiation to PDTC and ATC involves a number of additional mutations affecting other cell signaling pathways, such as *p53* and *Wnt*/β-*catenin*. Furthermore, *TERT* promoter mutations have been described in all the histological TC types, with a significantly higher prevalence in aggressive and undifferentiated tumors, indicating their role in TC progression [12,13,14,15,16,17,18,19,20]. Due to this evidence, we focused our attention on BRAF and TERT promoter mutations.

The aim of this study was to analyze a cohort of patients with LATC focusing on an overview about the different aspects linked with this diagnosis including their cytologic-histological correlation (when cytology was available), the histological type, and the risk for local recurrence, distant metastases, and overall survival.

## 2. Materials and Methods

A retrospective search was performed for all locally advanced thyroid cancers diagnosed over a 10-year period (January 2010 to December 2020) at the Fondazione Policlinico Universitario “Agostino Gemelli” in Rome, Italy. The institution’s electronic medical record system Armonia-Metafora, Italy (CU) was searched for thyroidectomy and thyroid lobectomy specimens during the same study period. All patient ages, gender, FNAC diagnoses, and follow-up surgical pathology information was recorded. All available pathology slides were reviewed. The majority of the thyroid nodules were evaluated and biopsied under ultrasound guidance by clinicians and radiologists. We received internal institutional ethical approval for this study (ID Study 3832).

### 2.1. Thyroid FNAC Specimens

All aspirations (with usually two passes performed for each thyroid lesion) were performed with 25 to 27 G needles. No rapid on-site assessment for adequacy of material was performed. All patients consented to their procedure. All FNAC specimens were processed using a ThinPrep 5000TM processor (Hologic Co., Marlborough, MA, USA). Prepared slides were fixed in 95% methanol and stained with a Papanicolaou stain. Any remaining material was stored in Preservcyt solution for potential ancillary studies.

Specimen adequacy was determined according to the Bethesda and British RCPath classification schemes [21,22,23,24,25,26,27,28,29,30,31]. The cytology cases were classified and diagnosed according to the new Italian Working Group SIAPEC-IAP classification [25,26]. All of the cases were re-evaluated and then re-classified according to The Bethesda System for Reporting Thyroid Cytology II (TBSRTC, 2017) [22]. For this retrospective study, analyses were conducted using TBSRTC terminology. This case series included the following distribution of diagnoses: 5.9% non-diagnostic (ND) including cystic cases; 77.8% benign lesions (BL); 3% atypia of undetermined significance/follicular lesions of undetermined significance (AUS/FLUS); 6.1% follicular neoplasms (FN); 2.2% suspicious for malignancy (SFM); and 5% positive for malignancy (PM) cases. All cytology and histology cases were reviewed by two cytopathologists whilst the re-classification, according to TBSRTC, was undertaken by one cytopathologist (EDR). Cases with an equivocal interpretation were subject to consensus review. The concordance between SIAPEC-IAP and TBSRTC classification systems was 95.9%.

### 2.2. Molecular Analysis for BRAF^V600E^ and TERT Mutation

DNA was extracted from both liquid-based cytology (LBC) stored aspirated material and paraffin-embedded tissue, according to our previous experience with the performance of ancillary techniques on thyroid samples [30,31,32,33,34,35]. *BRAF*^V600E^ mutational analysis was performed on DNA extracted from cytological and surgical specimens containing at least 70% tumor. Details of the molecular protocol employed have been previously published by our group [33,34,35]. Briefly, DNA was extracted from liquid-based cytology (LBC) samples stored in PreservCyt ^TM^ solution (Hologic, Marlborough, MA, USA) and from paraffin-embedded tissues. LBC sample was centrifugated; supernatant was discarded and cellular pellet processed. Pellet was incubated at 56 °C for three hours in 180 μL ATL lysis buffer and 20 μL Proteinase K (20 mg/mL) from QIAamp DNA mini kit (Qiagen, Hilden, Germany). For histological samples, 10 μm slide tissue was deparaffined and after ethanol treatment was incubated at 56 °C overnight in 180 μL ATL lysis buffer and 20 μL Proteinase K (20 mg/mL) from QIAamp DNA mini kit (Qiagen, Hilden, Germany). DNA was extracted following the manufacturer’s protocol and we spectrophotometrically assessed the quantity and quality of the DNA (A260, A260/280 ratio, spectrum 220–320 nm; Biochrom, Cambridge, UK) and by separation on an Agilent 2100 Bioanalyzer (Palo Alto, CA, USA). Low purity or insufficient DNA samples were extracted a second time. After a first amplification on a Rotor-Gene Q (QIAGEN), the mutational analysis of BRAF was achieved using “Anti-EGFR MoAb response (BRAF status)” (Diatech pharmacogenetics, REF: UP033) Kit by pyrosequencing via PyroMark Q96 ID system (Qiagen and Biotage, Uppsala, Sweden). Sensitivity of this method was at 5% in the CU laboratory [35]. The following sequence was analyzed: Exon 15: ACAGT/AGAAA. The percentage of disease specific cells for molecular analysis was at least 50% in all LBC samples and histological samples.

For *TERT*, genomic DNA was extracted from LBC samples stored in PreservCyt solution (Hologic, Marlborough, MA, USA) with a QIAamp DNA mini kit (Qiagen, Hilden, Germany), according to the manufacturer’s protocol. PCR was performed in 20 μL reactions containing genomic DNA (100 ng), 0.2 μmol/L of primers (forward 5′-CACCCGTCCTGCCCCTTCACCTT-3′ and reverse 5′-GGCTTCCCACGTGCGCAGCAGGA-3), and 2× PCRBIO HS Taq Mix (PCR Biosystems Inc., Wayne, PA, USA). PCR conditions were as follows: initial denaturation at 95 °C for 10 min followed by 35 cycles at 95 °C for 40 s, 62 °C for 40 s, and 72 °C for 40 s. *hTERT* promoter amplification was performed on an C1000 Touch Thermal Cycler (BioRad, Hercules, CA, USA). The yielded fragment was separated by electrophoresis on 2% agarose gel containing ethidium bromide and visualized by UV illumination. PCR product was treated with EXOSap (UBS, Sial, Rome, Italy), following the manufacturer’s protocol, and directly sequenced using a BigDye Terminator kit v3.1 (Applied Biosystem, Foster City, CA, USA) with forward and reverse primers in an ABI PRISM 3100 Genetic Analyser (Applied Biosystems, Foster City, CA, USA).

### 2.3. Histopathology Specimens

All surgical pathology specimens were fixed in 10% buffered formaldehyde, embedded in paraffin and 5 micron-thick sections, then stained with hematoxylin-eosin (H&E). The diagnosis of classical variant of papillary thyroid carcinoma (cPTC), different PTC variants, PDTC, MTC, ATC, and all the rare types of thyroid cancers were classified according to the WHO 2017 and following the criteria proposed in that WHO edition [36]. For the definition of tall cell variant (TCV) of PTC, we included cases of PTC with 30% or more TCV component. The histological diagnosis of noninvasive follicular thyroid neoplasm with papillary-like nuclear features (NIFTP) was rendered according to the criteria described by Nikiforov et al. [37]. All malignant cases were staged according to the seventh edition of the tumor-node-metastasis (TNM)-based staging system recommended by the American Joint Commission on Cancer (AJCC) [7]. Local recurrences and distant metastases were defined based on both the surgical procedures and clinical/radiological history of the patients.

According to the literature, the diagnosis of LATC is defined as locally advanced in the presence of an extra thyroid extension, e.g., when the surrounding structures such as the trachea, larynx, esophagus, and main blood vessels are invaded by cancer, and in presence of ETE and ENE, regardless of histotypes [12,13,14,15,16,17,18,19,20,21].

### 2.4. Statistical Analysis

Statistical analysis was performed using GraphPad-Prism 6 software (Graph Pad Software, San Diego, CA, USA) and PSPP version 1.4.1 (GNU Project, Free Software Foundation, Boston, MA, USA). Comparison of categorical variables was performed using the Chi-square statistic and the Fisher’s exact test. *p*-values less than 0.05 were considered statistically significant. We also performed the Kaplan Meier/Cox regression for survival analysis.

## 3. Results

Our 10-years thyroid surgical procedures included 20,543 thyroid cases including 8879 thyroid cancers. Our search was directed toward the identification of all the histological samples diagnosed as LATC, resulting in 65 cases. The patient demographics and clinical-pathologic features are described in Table 1. The series included 23 male (35%) and 42 female (65%) patients with an average age of 60.15 years (range 22–87 years). The mean size of thyroid nodules was 2.97 cm, ranging in size from 0.1 to 9 cm (Table 1), with a median dimension of 3 cm. A total of 42 out of 67 (64.6%) cases had a cytological evaluation, including the following distribution of diagnoses: 3 non-diagnostic, 0 benign; 2 AUS/FLUS; 3 FN/SFN; 5 SFM; and 29 M (Table 2). No statistical correlation was found with the available clinical-pathological data. For lymph node involvement, there were 28 cases (42.6%) that had central VI level involvement (N1a), whilst in 37 cases (57.4%), there were lateral cervical metastatic lymph nodes (N1b). The histological evaluation documented the presence of multifocality (more than 1 malignant foci) in 33 cases (50.8%), including 17 out of 33 with bilateral foci (i.e., 51.5% of multifocal cases and 26.15% of the global cohort).

Figure 1 specifies the correlation between thyroid tumor histotypes and multifocality. The distribution did not allow us to find any statistical correlation between these histotypes and multifocality. Table 2 shows the distribution for 65 cases with histopathological diagnoses. This series included 25 cases of cPTC, 2 I-FVPTC, 17 TCV, 1 hobnail variant of PTC, 1 solid variant of PTC, 7 columnar cell variants of PTC (CC-PTC), 2 follicular thyroid carcinomas (FTCs), 7 ATCs, and 1 MTC (Table 2). In Table 3, we highlight the cytologic-histological correlation for the 42 cases that had FNAs. Of note, the 2 AUS/FLUS cases on cytology were subsequently diagnosed as PTC on resection. The 3 FN/SFN cytology cases on follow-up were confirmed to be 1 PTC, 1 CC-PTC, and 1 FTC. The 5 SFM cases included 1 PTC, 1 I-FVPTC, and 3 TCV. The 29 malignant cytology cases reported pre-operatively were diagnosed on histopathology as 9 cPTC, 1 I-FVPTC, 1 solid variant of PTC, 3 hobnail variant of PTC, 5 CC-PTC, and 5 ATC. The 3 non-diagnostic cytology samples were histologically diagnosed as 2 cPTC and 1 ATC.

The most common anatomic sites to demonstrate carcinoma infiltration involved the inferior laryngeal nerve (45 cases; 69.2%) followed by the sternocleidomastoid muscles (16 cases; 24.6%), larynx (11 cases; 16.9%), trachea (11 cases; 16.9%), esophagus (8 cases; 12.3%), and jugular vein (5 cases; 7.6%). Tumor stage data revealed that there were 21 cases at stage I, 0 in stage II, 33 in stage III, and 7 with IVa and 4 with stage IVb. Figure 2 shows that the majority of cPTC cases belonged to stage I, and that the majority of aggressive variants of PTC (including TCV, CC-PTC, hobnail variant) were classified into stage III. ATC and the single MTC cases were allocated into stage IV. We also performed a Chi-square test, evaluating the association between tumor histotype and staging according to AJCC TNM 8th edition (2016), which yielded a *p* value of 0.013, thus showing a statistically significant difference with tumor stage among the various histotypes.

The results from molecular testing demonstrated that only three cases, all diagnosed as TCV, had *BRAF^V600E^* mutations and *TERT* genetic alterations. We correlated these genetic alterations with tumor size, multifocality, and lymph node metastases, but the limited number of mutated cases showed not statistically significant correlation.

We evaluated disease-free survival (DFS) and the overall free survival (OFS) with lymph node metastases (Figure 3a,b and Figure 4). As shown in Figure 3a,b, the DFS showed statistically significant differences among the various tumor histotypes using the Mantel-Cox test (*p* < 0.0001) and the Logrank test (*p* = 0.0127). Considering the different lymph nodal stations and DFS, we did not find any statistical correlation (Figure 4).

## 4. Discussion

Published data denote that the incidence of thyroid carcinoma has increased by nearly 300% in the last three decades [1,2,3,4,5,6,7,38]. The majority (>80%) of these thyroid cancers are well-differentiated carcinomas that tend to present with limited locoregional disease, leading to excellent long-term survival after surgical management [1,2,3,4,5,6,7,8,9,10,11,12]. Although the majority of thyroid cancers are detected in an early phase and are of small size, there has been an increasing number of larger tumors that manifest with locally advanced disease, mostly defined by ETE into nearby soft tissue, or even worse into adjacent organs [38,39,40,41,42,43,44,45,46,47]. Different articles found that ETE ranges from 5% to 34% [10,11,12,13]. Wang et al. reported an incidence of LATC of 4.1%, emphasizing that gross ETE predicts a worse prognosis [1]. We previously reported an incidence of LATC of 7.4% in our series that spanned 10 years in a single tertiary care referral center comprised mostly of high numbers of referred thyroid cancer patients to our institution, representing one of largest reference centers for thyroid disease/cancer in Italy. The most frequent site of ETE was infiltration of the recurrent laryngeal nerve, followed by invasion of the sternocleidomastoid muscles, larynx, and trachea.

The successful management of locally invasive thyroid cancer depends on a thorough understanding of the patterns of invasion, preoperative evaluation, and techniques of surgical resection and reconstruction. Moreover, the appropriate use of adjuvant therapy with radioactive iodine (RAI) and external beam radiation therapy (EBRT) is key to optimize management results. The optimal extent of surgery in cases of LATC remains controversial. Management of the disease requires a multidisciplinary approach, including endocrinology, nuclear medicine, oncology, endocrine surgery, pathology, and even general practice operating in different settings not always equipped with the appropriate services (such as specialized centers, general hospitals, and peripheral centers)

It is accepted that the definition of LATC is based mostly on the detection and extent of ETE [1,2,3,4,5,6,7,8,9,10,11,12,13,14]. To date, several studies have examined cases with LATC [1,2,3,4,5,6,7,8]. Patients presenting with LATC appear to be associated with a wide variety of cancer subtypes [38,48,49,50,51,52,53,54,55,56]. Many of these cases are aggressive variants of PT, PDTC, or ATC. These PTC variants include TCV, diffuse sclerosing variant (DSV), CC-PTC, as well as solid and hobnail variant of PTC, which have been reported to be of intermediate risk for recurrence [21], metastases, low response to radioiodine (RAI), and lower survival rate [21]. Our data confirmed that the majority of cases in this series were aggressive variants of PTC (Figure 5 and Figure 6). In fact, our series showed a prevalence of 25 cases of cPTC, 17 TCV, 7 CC-PTC, and also 7 ATC.

Moreover, as expected in these locally advanced cancers, many of these cases demonstrated multifocality of cancer, distributed in the same lobe or in both of them. However, we did not find any difference in multifocality between cPTC and aggressive variants of PTC.

FNAC was able to identify lymph nodal metastases and different histotypes in 90% and 5% of the cases, respectively. Even if some aggressive PTC cases can be recognized in cytological samples, TBSRTC does not advocate routinely diagnosing these variants [22]. The diagnosis of aggressive variants of PTC is mostly rendered on histological samples. In our series, all the cases had lymph node metastases including 57.4% of cases with pN1b disease. The majority of these pN1b cases had a histological diagnosis of an aggressive variant of PTC. This corroborates findings published by Limberg et al., showing that aggressive PTC variants even without invasive features, can be associated with nodal metastases [32]. In our series, the presence of LATC did not impact DFS or ODFS. The only relevant finding was represented by ETE. DFS showed a statistically significant difference among tumor histotypes using the Mantel–Cox test (*p* < 0.0001) and Logrank test (*p* = 0.0127). However, DFS had no statistical correlation with involvement of different lymph nodal stations.

Of note, our series was characterized mostly by large nodules with a mean size of 3.14 cm. This is in line with the thinking that small thyroid cancers are typically more indolent, as the same data have been confirmed by other authors [19,20,21,22]. Sugitani et al. demonstrated that risk of recurrence at 10 years following total thyroidectomy and neck dissection without RAI ablation was significantly higher in patients with pN1 disease with the largest metastatic LN > 3 cm (27%) than in patients with pN1 disease < 3 cm (11%). The risk of recurrence was significantly higher in patients with > 5 LN metastases (19%) than in those with <5 LN metastases (8%) [57].

In recent years, the identification of genetic alterations in thyroid carcinogenesis has shifted the focus on exploring the role of targeted therapies in thyroid cancer [48,49,50,51,52,53,54,55,56,57,58]. Mutations in the intracellular signaling pathways involving RAS, BRAF, and PI3K/Akt play a key role in thyroid tumor cell growth and survival [48,49,50,51,52,53,54,55,56,57].

Several molecular alterations have been found to be involved in the pathogenesis of thyroid cancer, especially PTC and its variants [38,48,49,50,51,52,53,54,55,56,57,58]. Among the genetic alterations, the diagnostic and prognostic value of *BRAF^V600E^* mutation and *TERT* promoter mutations have been well studied in PTC and other thyroid malignancies [50,51,52,53,54,55,56,57,58]. Whilst some studies have reported a correlation between *BRAF^V600E^* and more aggressive clinicopathological outcomes of cPTC, especially with respect to bilateral disease, ETE, and nodal involvement [48,49,50,51,52,53,54,55,56,57,58], we found only 3 mutated TCV cases. In this regard, one of the limitations is the number of cases, which is high for a single institution, but a larger and multi-center number of cases might provide additional information, including the evaluation of different managements.

## 5. Conclusions

In summary, although locally advanced thyroid cancers are uncommon, when encountered, they are challenging to manage. Our data suggest that LATC is often associated with an aggressive clinical course and should therefore accordingly be assertively managed. However, their surgical treatment is complex and may be associated with significant morbidity. Multidisciplinary planning is thus necessary to balance oncologic and functional outcomes. Additional studies are needed to better define LATC and establish more optimal treatment. Larger and multi-institutional series with a multi-disciplinary evaluation is necessary and it may represent the major limit of our single-tertiary thyroid center.

## Figures and Tables

**Figure 1 jpm-12-00221-f001:**
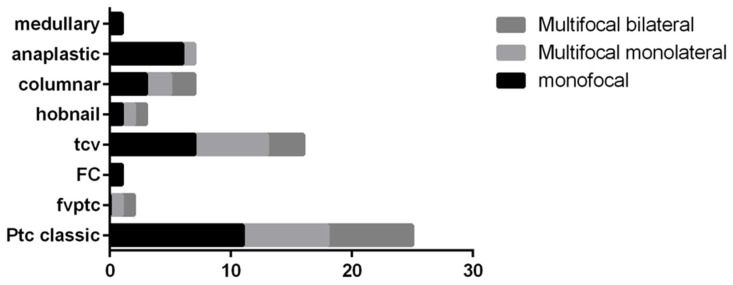
Distribution of thyroid cancer histotypes and multifocality; legend: PTC: papillary thyroid carcinoma; FC: follicular carcinoma; FVPTC: follicular variant of PTC; TCV: tall cell variant of PTC.

**Figure 2 jpm-12-00221-f002:**
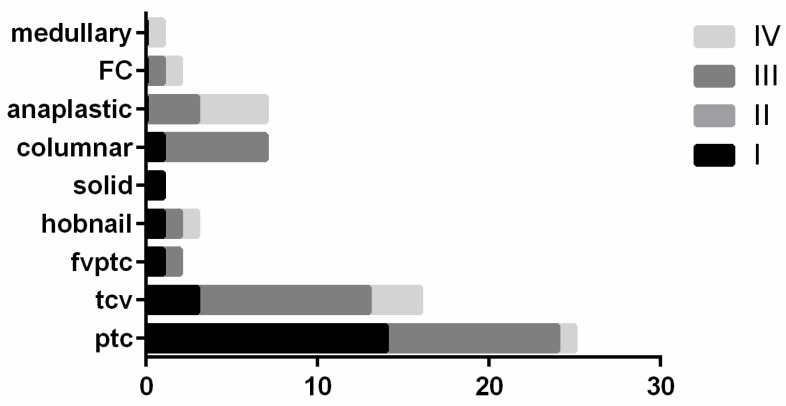
Distribution of thyroid cancer histotypes and stages of thyroid carcinomas.

**Figure 3 jpm-12-00221-f003:**
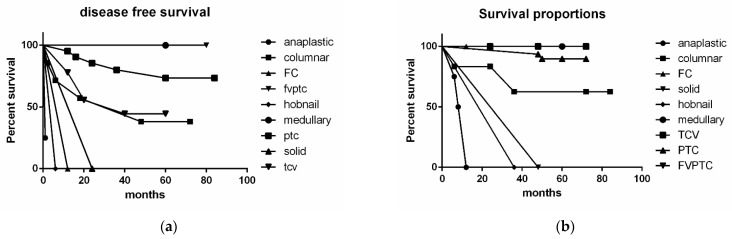
(**a**): Distribution of thyroid cancer histotypes and disease-free survival; (**b**): distribution of thyroid cancer histotypes and survival proportions.

**Figure 4 jpm-12-00221-f004:**
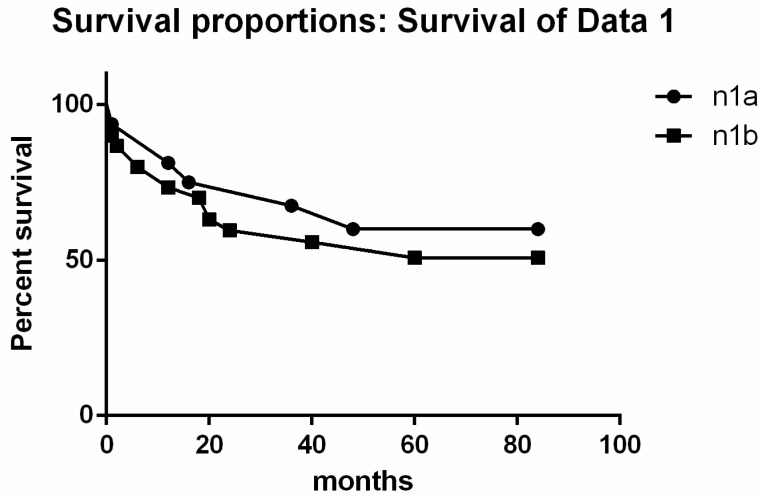
Survival proportions combined with the lymph-nodal metastases (IV level-N1a) and laterocervical (N1b).

**Figure 5 jpm-12-00221-f005:**
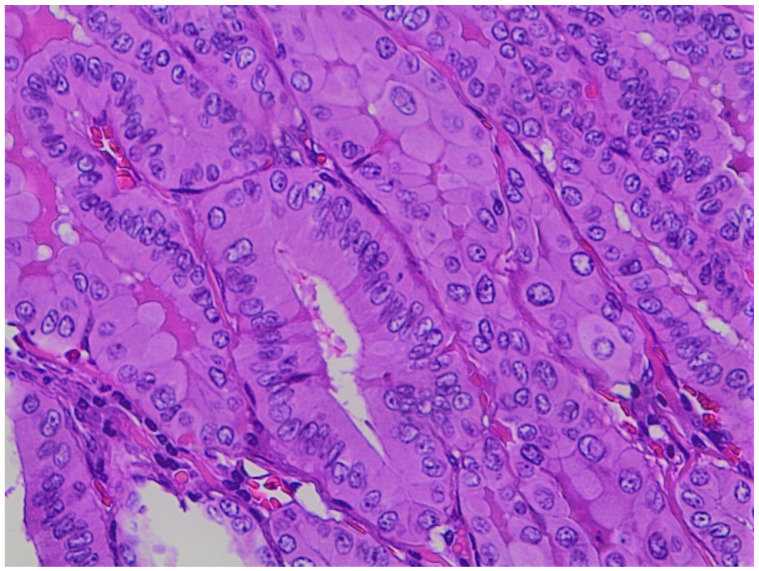
Details from a case of locally advanced thyroid carcinoma with tall cell variant of PTC component (200×, H&E).

**Figure 6 jpm-12-00221-f006:**
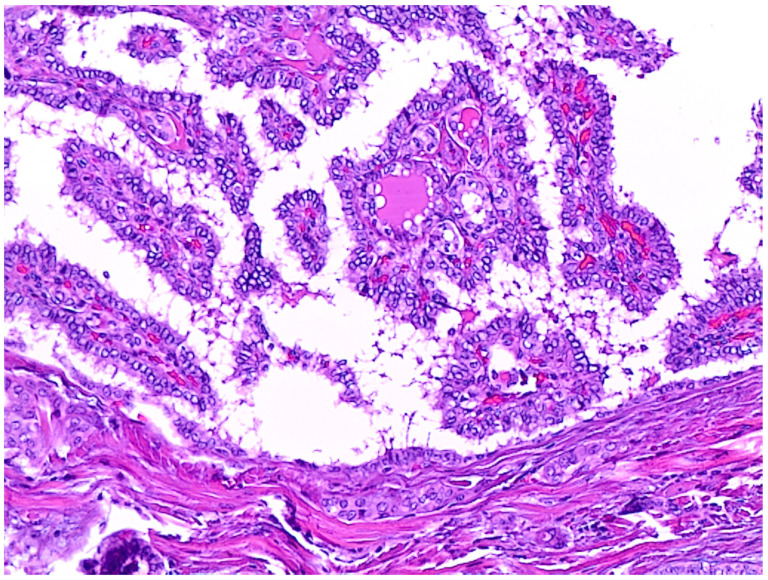
Details from a locally advanced thyroid carcinoma with a columnar cell variant of PTC component (200×, H&E).

**Table 1 jpm-12-00221-t001:** Summary of clinical-pathologic data.

Clinical-Pathological Features	Proportion (*n* = 65 Cases)
Age	
Mean	60.15 years
Median	60 years
Range	22–87 years
Gender	
Male	23 (35%)
Female	42 (65%)
Cytology diagnosis (*n* = 42 cases)	
Non-diagnostic	3 (7.1%)
Benign	0 (0%)
AUS/FLUS	2 (4.7%)
FN/SFN	3 (7.1%)
SFM	5 (11.9%)
Malignant	29 (69%)
Histopathology diagnosis	
Benign	0 (0%)
Malignant	65 (100%)
Lymph node involvement	
Central VI level	28 (42.6%)
Lateral cervical involvement	37 (57.45)
Multifocality	33

Legend: AUS/FLUS: atypia of undetermined significance/follicular lesion of undetermined significance; FN/SFN: follicular neoplasm/suspicious for follicular neoplasm; SFM: suspicious for malignancy.

**Table 2 jpm-12-00221-t002:** Histological diagnoses with pre-operative cytological diagnoses.

Histological Diagnoses	Number of Cases	Cases with FNAC
Classical PTC	25 (38.4%)	15 (60%)
I-FVPTC	2 (3%)	2 (100%)
TCV	17 (26.1%)	8 (47%)
CC-PTC	7 (10.7%)	6 (85.7%)
Hobnail PTC	3 (4.6%)	3 (100%)
Solid variant PTC	1 (1.5%)	1 (100%)
FTC	2 (3%)	1 (50%)
MTC	1 (1.5%)	0
ATC	7 (10.7%)	6 (85.7%)

FNAC: fine needle aspiration cytology ATC: anaplastic thyroid carcinoma; CC-PTC: columnar cell variant of PTC; PTC: Papillary thyroid carcinoma; FTC: follicular thyroid carcinoma; I-FVPTC: invasive follicular variant of PTC; MTC: medullary thyroid carcinoma; TCV: tall cell variant of PTC.

**Table 3 jpm-12-00221-t003:** Cyto-histological correlation.

Diagnoses	cPTC	I-FVPTC	FTC	TCV	SOLID PTC	HOBNAIL	CC-PTC	ATC
PM	9	1	0	5	1	3	5	5
SFM	1	1	0	3	0	0	0	0
SFN/FN	1	0	1	0	0	0	1	0
AUS/FLUS	2	0	0	0	0	0	0	0
BENIGN	0	0	0	0	0	0	0	0
ND	2	0	0	0	0	0	0	1

Legend: AUS/FLUS: atypia of undetermined significance/follicular lesion of undetermined significance; SFN/FN: follicular neoplasm/suspicious for follicular neoplasm; SFM: suspicious for malignancy; cPTC: classic papillary thyroid carcinoma, I-FVPTC: invasive follicular variant of PTC, FTC: follicular thyroid carcinoma; CC-PTC: columnar cell variant of PTC, ATC: anaplastic thyroid carcinoma, MTC: medullary thyroid carcinoma, ND = non diagnostic.

## Data Availability

Not applicable.

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
