# Peer review of "Does Locally Advanced Thyroid Cancer Have Different Features? Results from a Single Academic Center"

_jpm, 2022, doi:10.3390/jpm12020221_

Round 1

Reviewer 1 Report

This is an interesting study reporting data of locally advanced TC reported in a high-volume institution. The aim is clear, the methods are correct, the results are clearly reported, the conclusions of the study are supported by the results.

In my opinion the paper can be accepted after a minor revision:

  1. please define in the methods how were assessed "local recurrence" and "distant metastases".
  2. please revise the title of figure 4.

Author Response

We added the requested specification in the method section.

we noticed that Figure 4 had a title in the manuscript line 207-8 pag 7

Reviewer 2 Report

I have the following issues that require further consideration.

  1. In introduction, the authors are suggested to be more specific and clearly set the rationale and novelty of their study- what is already known and what does this study add to the current literature. Their number of cases? The combined cytologic-histological evaluation of samples? The data presented about survival analysis? Please clarify.
  2. The definition of locally advanced thyroid cancer (LATC) should be clearer, stricter and based on specific published references, because readers who are not familiar with this terminology can become confused. If extrathyroidal extension (ETE) and extranodal extension (ENE) are not enough to define them and certain histological findings are also suggestive of this group of thyroid cancer, which types of thyroid cancer are NOT described under this definition (LATC)? Please clarify.
  3. Following the previous comment, apart from the known main types of well- and poorly -differentiated thyroid cancers (papillary, follicular, Hurthle cell, Medullary, Anaplastic thyroid cancer), some particular histopathological types are also given as acronyms throughout the text. In my opinion, a brief summary of the classification of thyroid cancer could be helpful.
  4. Before stating the aim of their study, some data regarding the current management of LATC (surgical/ pharmaceutical strategies) and related prognosis would certainly be helpful as well.
  5. The authors should also try to cover any potential underlying mechanisms, including the signaling pathways and the markers that have been so far involved in LATC. Which is the role and why did they choose to test BRAF and TERT mutations?
  6. The inclusion and exclusion criteria of the study participants/ samples and the stages/phases of their procedures should be given in more detail, preferably in a structured way. A flow chart would be helpful (how many patients/ samples with thyroid cancer were identified in total over the 10-year period, how many of them were defined as LATC, and how many of the experimental procedures were followed in the 65 samples of the study (at what percentage) – FNA cytology, histopathology, mutational analysis, surgical management etc These details would be necessary to avoid confusion.
  7. Details of the molecular protocol employed for the BRAFV600E mutational analysis are suggested to be briefly presented as well (not only reference to previous published work).
  8. In statistical analysis, additional methods used by the authors have to be added (Kaplan meier/ cox regression for survival analysis etc).
  9. Analysis of data seems to be a bit problematic. (a) There are too many thyroid cancer subtypes (based on cytology, histology), unclear if some of them should be grouped together with others or not [classical variant of papillary thyroid carcinoma (cPTC), different PTC variants, poorly-differentiated TC (PDTC), MTC and ATC], [AUS/FLUS: Atypia of Undetermined Significance/Follicular Lesion of Undetermined Significance; FN/SFN: Follicular Neoplasm/Suspicious for Follicular Neoplasm; SFM: Suspicious for Malignancy, malignancy (PM)], (b) a definition of multifocality is missing, (c) not all p of comparisons are given (so unclear whether there is statistical significance or not; authors do present only percentages), (d) I am not sure if there is any relation between thyroid cancer histotypes and multifocality or staging and which is the most frequent presentation in patients with LATC (based on figures 1 and 2 ), (e) not sure which is the significance of tables 2 and 3, (f) Figures 3a and 3b do not easily convey their message; and (g) in some cases conclusions are not clear, for example it’s unclear whether there is an association between LATC and poor prognosis or not: it is concluded that “the majority of cases in this series were aggressive variants” but there is a wide range of disease survival according to the particular diagnosis of LATC and significance is lost after the presence of lymphnodal metastases is taken into account - authors also declared: “In our series, the presence of LATC did not impact DFS or ODFS. The only relevant finding was represented by ETE. DFS showed a statistically significant difference among tumor histotypes. …., DFS had no statistical correlation with involvement of different lymph nodal stations”. Please clarify / comment on all these comments.
  10. Discussion/ conclusions need to be increased in length. The authors are strongly recommended to emphasize on their own main findings/ conclusions of their current study, highlight, discuss them and try to explain any discrepancies with the literature. Moreover, they should also describe the limitations of their study. Yet, the number of references needs to be reduced and authors are suggested to maintain in their list only the most relevant and recent publications.
  11. Figures 5 and 6 –which present histopathological findings-, can be of some educative value but do not seem to add too much in the interpretation of this study’s findings and could be omitted; if finally used, at least arrows that show what the legend of the figure describes should be used.
  12. Several grammatical/ syntax errors exist throughout the text. The authors should carefully revise their manuscript with the help of a native English speaker.

Author Response

1-we clarified the objects which are mostly an overview of all the different aspects related to these LATCS

2-we clarified as requested

3-we added a note in the material about the criteria used in line with the WHO 2017

4-We added and specified some details about the management of LATCs

5-We added some sentences about molecular mechanisms

6-We specified as requested in the results. WE did not create a chart, mostly becuase we included all the cases with a histological diagnosis of LATC. However we specified the number of thyroidectomies and the number of thyroid cancers too

7-We added the molecular protocol as suggested.

8-We added as requested the additional statistics

9-a)We need to maintain all the subtypes of histological and cytological diagnoses as for our purpose of link the possible LATC with different histotypes; b) we added the definition of multifocality in the result section, when we introduced the figure of multifoci; C) we presented % for all the data and p for the significant values; d) we expanded the comments in the discussion section

10-We added some more comments about our series including more comparison with the literature. We reduced the number of references

11-The figures show the 100% evidence of the variant, so we avoided arrows because it is a uniform sinilar pattern in both of them

12) we revised the paper as requested